# Kinase Inhibitors and Kinase-Targeted Cancer Therapies: Recent Advances and Future Perspectives

**DOI:** 10.3390/ijms25105489

**Published:** 2024-05-17

**Authors:** Jiahao Li, Chen Gong, Haiting Zhou, Junxia Liu, Xiaohui Xia, Wentao Ha, Yizhi Jiang, Qingxu Liu, Huihua Xiong

**Affiliations:** Department of Oncology, Tongji Hospital, Tongji Medical College, Huazhong University of Science and Technology, Wuhan 430030, China; m202276284@hust.edu.cn (J.L.);

**Keywords:** kinase activity inhibitors, degraders, antibodies, nanobodies, protein–kinase interaction inhibitors, cancer therapy

## Abstract

Over 120 small-molecule kinase inhibitors (SMKIs) have been approved worldwide for treating various diseases, with nearly 70 FDA approvals specifically for cancer treatment, focusing on targets like the epidermal growth factor receptor (EGFR) family. Kinase-targeted strategies encompass monoclonal antibodies and their derivatives, such as nanobodies and peptides, along with innovative approaches like the use of kinase degraders and protein kinase interaction inhibitors, which have recently demonstrated clinical progress and potential in overcoming resistance. Nevertheless, kinase-targeted strategies encounter significant hurdles, including drug resistance, which greatly impacts the clinical benefits for cancer patients, as well as concerning toxicity when combined with immunotherapy, which restricts the full utilization of current treatment modalities. Despite these challenges, the development of kinase inhibitors remains highly promising. The extensively studied tyrosine kinase family has 70% of its targets in various stages of development, while 30% of the kinase family remains inadequately explored. Computational technologies play a vital role in accelerating the development of novel kinase inhibitors and repurposing existing drugs. Recent FDA-approved SMKIs underscore the importance of blood–brain barrier permeability for long-term patient benefits. This review provides a comprehensive summary of recent FDA-approved SMKIs based on their mechanisms of action and targets. We summarize the latest developments in potential new targets and explore emerging kinase inhibition strategies from a clinical perspective. Lastly, we outline current obstacles and future prospects in kinase inhibition.

## 1. Introduction

Kinases are enzymes that facilitate the transfer of high-energy phosphate groups from ATP to specific substrates, resulting in the generation of ADP and substrates phosphorylated at specific sites. The human kinome comprises approximately 560 protein kinases, with about 500 being eukaryotic protein kinases sharing sequence similarities [1]. In addition, there are over 60 atypical kinases, including lipid kinases, etc., that do not exhibit sequence similarity [1]. Subcellular localization studies have revealed that roughly 50.2% of kinases are predominantly found in the cytoplasm, including tyrosine and serine/threonine kinases [2]. Approximately 16.4% are located in the cell nucleus, and 10.3% are on the plasma membrane, including receptor tyrosine kinases (RTKs) [2]. The phosphorylation of substrates by kinases can influence enzyme activity or alter a kinase’s interaction with its protein targets [3], thereby regulating cellular functions like growth, differentiation, proliferation, apoptosis, inflammation, and metabolism [4,5]. This regulatory role extends to initiating primary oncogenic transformation and maintaining tumors [6].

The development of tumor treatment strategies targeting protein kinases has been a subject of interest since the 1980s [7]. The types of kinase inhibitors used in tumor treatment can be categorized into small-molecule kinase inhibitors (SMKIs) and larger molecules such as monoclonal antibodies (mAbs), nanobodies, and peptides [8,9,10,11]. SMKIs target the highly conserved ATP-binding pocket of protein kinases, thus constituting kinase inhibitors or kinase activity inhibitors that target the deep structural domain [12]. The advent of targeted therapy driven by kinase inhibitors began with the approval of *Imatinib* in 2001, marking a significant milestone in cancer treatment [13]. Currently, kinase inhibitors represent the largest and most approved category of drugs among all tumor treatment medications [14]. Over 120 kinase inhibitors have been approved worldwide, primarily for cancer and autoimmune diseases. In the past six years, the FDA has approved 33 kinase inhibitors or mAbs for cancer treatment, close to half of all kinase inhibitors for tumors. The EGFR/HER family represents the most mature kinase target to date, with 18 kinase inhibitors developed that target this family, including mAbs and derivatives. The latest SMKIs targeting the EGFR/HER family are primarily designed to overcome resistance caused by mutations, such as EGFR L858R, T790M, and C797S [15].

Antibodies that target RTKs can bind to the transmembrane extracellular segments of these kinases, inhibiting their activation and blocking signal transduction. Trastuzumab, approved by the U.S. Food and Drug Administration (FDA) in 1998 for HER2-positive metastatic breast cancer, marked the beginning of large-molecule targeted therapy against kinases [16]. Due to the limited tissue penetrability of full antibodies, researchers have developed mAb derivatives like nanobodies and peptides, which are believed to have enhanced tissue penetration capabilities and improved bioavailability. Nanobodies, derived from the antigen recognition domain of camelid heavy-chain antibodies, possess these properties because they are much smaller than traditional antibodies [17].

New strategies for targeting kinases include kinase degraders and protein kinase interactors. Kinase degraders encompass proteolysis-targeting chimeras (PROTACs), lysosome-targeting chimeras (LYTACs), dTAGs, hydrophobic tagging, and “Trim-Away” technologies. On the other hand, protein kinase interactors build upon the concept of protein–protein interaction (PPI) inhibitors, mainly disrupting the interaction between kinases and substrates without affecting enzymatic activity [18]. Peptides are considered promising candidates for protein kinase interactors due to their more flexible conformations and higher selectivity [19]. These emerging strategies show promise in developing new targets and overcoming drug-resistant mutations, with some already progressing to clinical trials.

Kinase inhibitors continue to face challenges and dilemmas in the current landscape of cancer treatment. One of the major hurdles is the development of acquired resistance to SMKIs, which hinders long-term patient benefits. The prolonged use of SMKIs inevitably leads to acquired resistance, primarily through on-target mutations and off-target resistance mechanisms. On-target mutations typically occur in key regions of kinases, such as the ATP-binding pocket, solvent front, gatekeeper, and xDFG regions, directly impacting drug-target binding [20]. On the other hand, off-target resistance is mainly driven by bypass pathway activation, where downstream pathways are activated to compensate for the inhibition of the target kinase, leading to resistance [21]. To combat acquired resistance in SMKIs targeting the EGFR/HER family, third-generation SMKIs have been developed to target different sites of EGFR mutations. Furthermore, various bypass activation pathways, like ALK, RET, and ROS1, contribute to resistance beyond the EGFR, prompting the development of inhibitors targeting these new variant targets, with several inhibitors receiving approval in recent years.

Recently approved SMKIs have demonstrated efficient blood–brain barrier permeability, potentially enhancing their efficacy in patients with late-stage brain metastases. This suggests a promising future direction for the development of kinase inhibitors. Additionally, emerging targeted kinase strategies have shown encouraging results in overcoming point mutations in preclinical settings. On the other hand, immune checkpoint inhibitors (ICIs) have significantly improved the prognosis for certain cancer patients. However, the combined use of kinase inhibitors and ICIs is currently limited to a few tumors due to the heightened toxicity associated with this approach, sparking ongoing debates in the field. It is evident that there is still ample opportunity for further advancements in kinase inhibition, as only a fraction of the kinase family has been targeted, with approximately 30% of kinases lacking sufficient research. With the rapid progress in computer technologies, such as artificial intelligence and machine learning, the development of kinase inhibitors can be expedited through molecular design, computer screening, the prediction of protein–protein interactions, and drug repurposing. This review aims to provide a comprehensive summary of the mechanisms and targets of recently FDA-approved SMKIs, as well as to discuss novel kinase inhibition strategies, including nanobodies, kinase degraders, and protein–kinase interaction inhibitors (PKIIs). Finally, the review will address the current challenges and future directions for the development of kinase inhibitors.

## 2. Types of Kinase Inhibitors

### 2.1. Mechanism of Action and Classification of SMKIs

#### 2.1.1. Structural Features of Kinase

The majority of the aforementioned 33 kinase inhibitors recently approved by the FDA for cancer treatment are SMKIs, with additional detailed information available in Table 1. The common catalytic domain of protein kinases is depicted in Figure 1a. The ATP-binding sites of these domains display structural similarity, allowing them to be recognized and bound by various common hinge-binding scaffolds like (iso)quinoline, pyrimidine, quinazoline, and pyrrolo or imidazo triazines [5,22,23,24,25,26]. The kinase domain consists of a smaller N-terminal lobe primarily made up of β sheets and a conserved C helix, along with a larger, mostly helical C-terminal lobe containing many residues phosphorylated during kinase activation [27]. Two lobes are connected by a hinge region [27]. The G-rich loop plays a role in catalyzing phosphoryl transfer reactions within the catalytic core of protein kinases. The αC–β4 loop moves outward when the kinase is inactive and inward when the kinase is active. HRD is directly involved in autophosphorylation, controlling the activation state of protein kinases. ATP binds in the folding cleft between the lobes, interacting with the hinge through conserved hydrogen bonds [12]. Moreover, the Asp–Phe–Gly (DFG) motif is a conserved motif that coordinates with a magnesium ion involved in catalysis, and its movement is crucial for adopting the active conformation [5]. A-loop refers to the activation loop, with its N-terminus determined by the DFG motif. The “DFG-in” conformation represents the active state, while “DFG-out” indicates the inactive state. The αC helix plays a vital role in the conformational dynamics of protein kinases, tightly packing against the rest of the kinase domain in the “αC-in” state. Some kinases exhibit a loosely attached αC helix in their inactive state, known as “αC-out” [27]. Only kinases in the “DFG-in” and “αC-in” conformations are considered active. Any displacement of the αC helix or DFG motif from the “in” position disrupts their interactions, opening pockets inside and around the ATP-binding site and facilitating drug binding. Current research suggests that the active conformation of kinases includes the regulatory R-spine, characterized by a stacked configuration of four side chains [12]. Additionally, variable DFG-D or αC-helix conformations have been observed [28]. There are regulatory sites near the ATP-binding pocket [29]. The development of allosteric inhibitors can modify kinase conformations by targeting specific sites [29]. In this review, SMKIs are classified into three classes and five types according to their mode of action and binding site positions (Figure 1b). The varied conformations of protein kinases and kinase inhibitor complexes, along with multi-site binding, have prompted more intricate classification methods resulting from more complicated mechanisms of action, although these are beyond the scope of this review, referring to Roskoski et al. [28] and Laufkotter et al. [30].

Figure 1 shows a schematic diagram of the general structure of kinases and the types of kinase inhibitors based on the mechanism of action.

#### 2.1.2. ATP-Competitive Inhibitors

The competitive inhibition of ATP binding is currently the most established method of kinase inhibition, and most kinase inhibitors fall into this category. ATP-competitive inhibitors are classified as Type I or Type II [14]. Type I kinase inhibitors, the largest group of inhibitors, competitively inhibit ATP by binding to amino acid residues in the ATP-binding pocket or hinge region; these inhibitors are characterized by their ability to adopt the DFG-in conformation. Further subdivision based on the αC helix can be found in a study by Arter et al. [12]. Recently approved SMKIs in this category include *capivasertib, capmatinib, selpercatinib, entrectinib, erdafitinib,* and *gilteritinib* [31,32,33,34,35,36].

On the other hand, Type II inhibitors bind to the inactive DFG-out conformation of kinases, regardless of αC-in or αC-out. Because the inactive protein kinase conformation exhibits greater structural variability than the conserved active conformation bound by Type I inhibitors, it is speculated that Type II inhibitors are more selective [37]. *Quizartinib, pirtobrutinib*, and *repotrectinib* are all Type II ATP-competitive inhibitors [38,39,40].

#### 2.1.3. Allosteric Inhibitors

Allosteric kinase inhibitors, competitive inhibitors that do not compete with ATP, bind to sites outside the kinase ATP-binding pocket. This means they do not directly interfere with the hinge region of the ATP-binding site. Examples of these inhibitors include *binimetinib* and *asciminib* [41,42]. Depending on the location of the allosteric sites in relation to the ATP-binding pocket, these inhibitors are classified as Type III or Type IV. Type III inhibitors bind to sites near the ATP-binding pocket and may have higher selectivity than Type I or II inhibitors. Examples of Type III inhibitors are *pexidartinib* and *selumetinib* [43,44]. On the other hand, Type IV inhibitors bind to sites that are distinct from the ATP-binding pocket.

#### 2.1.4. Covalent Inhibitors

Type V inhibitors, also known as covalent inhibitors, typically form a covalent bond with the thiol groups of cysteine or lysine residues at the ATP site, effectively blocking enzyme activity either reversibly or irreversibly. Several covalent kinase inhibitors, such as *futibatinib*, *mobocertinib*, *zanubrutinib*, and *dacomitinib*, have been approved [45,46,47,48]. Irreversible covalent binding can enhance drug selectivity, which can aid in the development of selective inhibitors and overcoming resistance [48,49]. However, irreversible kinase binding has introduced new challenges, including increased side effects, unpredictable toxicity, and immunogenicity from covalent complexes [50,51,52]. Despite the effectiveness of second-generation EGFR inhibitors like *afatinib* and *dacomitinib*, they also inhibit the wild-type EGFR, leading to severe side effects. Reversible covalent inhibitors may provide a solution to these challenges. Reversible covalent inhibitors do not permanently bind to kinases, allowing for release from off-target proteins and reducing the risk of immune system adverse activation and off-target toxicity [53]. *Rilzabrutinib*, a reversible covalent Bruton’s tyrosine kinase (BTK) inhibitor, has promising potential in immune thrombocytopenia [54]. Its slow dissociation from BTK results in a prolonged duration of action, leading to a sustained target occupancy that helps minimize the adverse reactions caused by off-target nonspecific binding [55].

#### 2.1.5. Type Selection for the Development of SMKIs

The strategy of multi-kinase inhibition leads to toxic side effects, and its high selectivity can easily lead to resistance. Developing covalent inhibitors with enhanced specificity, potency, and safety is crucial for expanding their therapeutic applications. However, covalent inhibitors are not necessarily superior to ATP-competitive inhibitors, especially considering the relatively mature system of ATP-competitive inhibitor development. Several approved irreversible covalent BTK inhibitors such as *ibrutinib*, *acalabrutinib*, and *zanubrutinib* have demonstrated off-target side effects, suboptimal pharmacokinetics for oral administration, and limitations related to BTK C481 mutation resistance [39]. Conversely, the non-covalent, reversible, selective BTK inhibitor *pirtobrutinib* has demonstrated efficacy higher than the aforementioned covalent inhibitors and is effective against both wild-type BTK and various BTK C481 substitution mutants [56]. This suggests that different targets, conformations, and mutant variants may have their own suitable types of inhibitors. Additionally, multi-targeted inhibitors may have different mechanisms of action against different targets. For example, although *sunitinib* binds to some kinases in a Type I conformation, it is a Type II inhibitor of KIT [57]. Therefore, there is no definitive answer to which mode of action for SMKIs is currently the best.

**Table 1 ijms-25-05489-t001:** FDA-approved kinase inhibitors and monoclonal antibodies from 2018 to 1 March 2024.

Compound Name	First Approved	Target	Type	Reference
Quizartinib	2023	FLT3 inhibitors	Type II	[58]
Capivasertib	2023	AKT-1, AKT-2, AKT-3	Type I	[59]
Pirtobrutinib	2023	BTK C481S inhibitor	Type II	[39]
Repotrectinib	2023	ALK, ROS1, TRKA, TRKC	Type II	[40]
Fruquintinib	2023	VEGFR1, VEGFR2, VEGFR3	Type I/II	[60]
Futibatinib	2022	FGFRs	Type V	[45]
Tepotinib	2021	c-Met	Type I	[61]
Amivantamab	2021	EGFR, c-Met		[62]
Asciminib	2021	BCR–ABL	Type IV	[42]
Infigratinib	2021	FGFR1, FGFR2, FGFR3, FGFR4	Type I/II	[63]
Mobocertinib	2021	EGFR exon 20, HER2 exon 20	Type V	[46]
Umbralisib	2021	CK1ε, PI3Kδ	Type I/II	[64]
Tucatinib	2020	HER2	Type I/II	[65]
Avapritinib	2020	PDGFRα, c-Kit	Type I	[66]
Capmatinib	2020	c-Met	Type I	[32]
Margetuximab	2020	HER2	Monoclonal antibody	[67]
Pemigatinib	2020	FGFR1, FGFR2, FGFR3	Type I/II	[68]
Pralsetinib	2020	RET	Type I	[69]
Selpercatinib	2020	RET	Type I	[33]
Selumetinib	2020	MEK1, MEK2	Type III	[44]
Ripretinib	2020	EGFR, PDGFRα, c-Kit	TypeII	[57]
Alpelisib	2019	PI3Kα	Type I/II	[70]
Entrectinib	2019	ALK, ROS1, TRKA, TRKB, TRKC	Type I	[34]
Erdafitinib	2019	FGFR1, FGFR2, FGFR3, FGFR4	Type I	[71]
Pexidartinib	2019	CSF-1R, FLT3, c-Kit	Type III	[72]
Zanubrutinib	2019	BTK	Type V	[47]
Duvelisib	2018	PI3Kγ, PI3Kδ	Type I/II	[73]
Encorafenib	2018	BRAF	Type I/II	[74]
Gilteritinib	2018	AXL, FLT3	Type I	[36]
Dacomitinib	2018	EGFR, EGFRL858R, EGFR-Ex19del, HER2, HER4	Type V	[48]
Lorlatinib	2018	ALK, ROS1	Type I/II	[75]
Binimetinib	2018	MEK1, MEK2	TypeIII/IV	[76]
Larotrectinib	2018	TRKA, TRKB, TRKC	Type I/II	[77]

### 2.2. Monoclonal Antibodies

mAbs primarily inhibit signal transduction via the binding of the antibody’s Fab segment to the receptor’s extracellular domain (Figure 1c). The most successful mAb targets for kinases are the EGFR/HER family, with a total of seven EGFR/HER-family-targeting mAbs approved. Trastuzumab was the earliest of this type of mAb to be approved, whereas the FDA approved ramucirumab (targeting VEGFR) and olaratumab (targeting PDGFR) in 2014 and 2016, respectively. In recent years, few mAbs targeting RTKs have been approved, posing significant challenges for the clinical development of mAbs targeting RTKs. The recent approval of amivantamab, a bispecific mAb targeting both EGFR and c-MET, seems to bring hope for mAb development [78]. EGFR exon20ins confer resistance to traditional EGFR TKIs. Amivantamab targets both EGFR exon20ins and MET, leading to signal blockade and degradation. Therefore, accelerated FDA approval was granted in May 2021 for treating locally advanced or metastatic NSCLC in adult patients carrying EGFR exon 20 mutations [78,79]. This indicates the potential of RTK mAbs to simultaneously target on-target resistant mutations and inhibit bypass activation pathways.

## 3. FDA Approved Kinase Inhibitors in Oncology in 2018–2024

Several articles have extensively discussed FDA- or globally approved kinase inhibitors [5,7,80]. Roskoski et al. have been summarizing the physicochemical properties of FDA-approved kinase inhibitors, including lipophilic efficiency and ligand efficiency, since 2019 [28,37,81,82,83,84]. In addition, there are websites regularly updating approved or clinically tested kinase inhibitors (https://brimr.org/protein-kinase-inhibitors/; https://www.icoa.fr/pkidb/, accessed on 1 March 2024). For SMKIs approved by the FDA between 2018 and 1 March 2024, we provide a summary based on timelines and targets in Figure 2. Our analysis primarily focuses on targets or mutants that were previously unapproved, including TRK, FGFRs, CSF-1R, and AKT, among others. Furthermore, we further elucidate the resistant mutations and mechanisms of these approved targets.

### 3.1. Tropomyosin Receptor Kinase (TRK)

The tropomyosin receptor kinases A, B, and C (TRKA, TRKB, and TRKC) belong to the RTK family. These receptors play a crucial role in regulating cell proliferation, differentiation, metabolism, and apoptosis by phosphorylating downstream targets. The transmembrane fusion of NTRK genes often leads to the absence of extracellular domains, making common targeted therapies like monoclonal antibody treatments ineffective. Therefore, the primary strategy for targeting NTRK fusion proteins involves using selective SMKIs of TRK. *Larotrectinib* and *entrectinib*, both FDA-approved in 2018 and 2019, respectively, target TRK effectively [34,85].

*Larotrectinib* is an ATP-competitive inhibitor of TRKA, TRKB, and TRKC [85], showing promising results in clinical trials involving 55 patients with various types of TRK fusion-positive tumors, indicating an overall response rate (ORR) of 75% [86]. Consequently, in November 2018, it was approved for metastatic solid tumors in both adults and children. On the other hand, *entrectinib* is a potent Type I inhibitor targeting TrkA/B/C, ROS1, and ALK with impressive response rates in patients with advanced or metastatic NTRK fusion-positive solid tumors [86]. The clinical trials STARTRK-1 and STARTRK-2 demonstrated that among 54 patients with advanced or metastatic NTRK fusion-positive solid tumors, 31 patients (57%) achieved objective responses, with 4 patients (7%) achieving complete responses (CRs), and the median duration of response (DOR) was 10 months [87].

However, currently approved NTRK inhibitors still face challenges in drug resistance. The median DOR for the NTRK fusion subgroups to *larotrectinib* was 8.3 months, and for *entrectinib*, it was 10.5 months [87]. This is believed to be associated with point mutations in the kinase domain of TRK, including solvent front mutations, gatekeeper mutations, and xDFG motif mutations, such as TRKA G667C/A/S, TRKB G709C/A/S, and TRKC G696C/A/S, as well as the aberrant activation of bypass signaling pathways in TRK, such as acquired BRAF V600E and KRAS G12D mutations, and MET amplification, leading to MAPK pathway activation [87,88]. To overcome acquired resistance mutations, *selitrectinib* and *repotrectinib* are two representative second-generation TRK inhibitors. They are capable of effectively inhibiting xDFG alterations both in vitro and in vivo. *Selitrectinib* is currently undergoing Phase II clinical studies, and *repotrectinib* is in Phase III trials. However, these drugs exhibit significant off-target inhibitory effects, highlighting the ongoing necessity for the development of highly selective Type II TRK inhibitors [89,90].

### 3.2. Fibroblast Growth Factor Receptors (FGFRs)

FGFRs 1–4 are part of the RTK family, including an intracellular tyrosine kinase structure [91]. Amplifications, mutations, and rearrangements of FGFRs have been frequently reported to drive tumor initiation and progression, especially in urothelial carcinoma (UC). In early-stage UC, over half of the cases show FGFR3 alterations [92], while around 20% of advanced/metastatic UC cases have FGFR3 mutations. Various FGFR kinase inhibitors have shown promise in targeting both cancer cells and the tumor microenvironment (TME). Several FGFR inhibitors have already been approved for metastatic urothelial carcinoma and cholangiocarcinoma.

*Erdafitinib* is a pan-FGFR Type I inhibitor, targeting FGFR1–4 [35]. A Phase II study (NCT02365597) on *erdafitinib* demonstrated an ORR of 42% (comprising 3% complete response and 39% partial response) in 96 patients with metastatic or unresectable urothelial carcinoma and FGFR alterations. Notably, in patients previously treated with ICIs (n = 21), a confirmed ORR of 70% was observed. *Pemigatinib* and *infigratinib* are also pan-FGFR ATP-competitive inhibitors. *Pemigatinib* selectively inhibits FGFR1–3, showing weaker activity against FGFR4 and other non-FGFR4 subtypes [68]. The FIGHT-202 trial (NCT02924376) assessed *pemigatinib* in 107 patients with locally advanced unresectable or metastatic cholangiocarcinoma, reporting an ORR of 36% (including 2.8% complete response and 33% partial response) [93]. *Infigratinib* inhibits FGFRs 1–4. The CBGJ398X2204 trial (NCT02150967) studied *infigratinib* in 108 unresectable locally advanced or metastatic cholangiocarcinoma patients with FGFR2 fusion or rearrangement, showing an ORR of 23% [94]. Both *infigratinib* and *pemigatinib* have received approval for cholangiocarcinoma treatment. The FDA approved the irreversible covalent inhibitor *futibatinib* in 2022, with IC50 values for FGFRs 1–4 all below 4 nM. The TAS-120-101 trial (NCT02052778) evaluated *futibatinib* in 103 unresectable cholangiocarcinoma patients, including FGFR2 fusion or rearrangement patients, reporting an ORR of 42% and a 42% PR [95].

Resistance mutations of FGFRs are often polyclonal, especially in FGFR2 mutations, where patients can acquire multiple mutations during treatment. Secondary FGFR2 mutations are polyclonal in 50–63% of patients with intrahepatic cholangiocarcinoma. Mutations such as FGFR2 V564F/I/L in gatekeeper, FGFR2 N549D/K and E565A in molecular-brake, and FGFR2 L617F/V in DFG-latc are all associated with resistance to *infigratinib* and *pemigatini* [96]. The FGFR2 V564F/L and N549D/K mutations also confer resistance to erdafitinib, while FGFR2 V564F/L leads to resistance to *futibatinib*. Additionally, in some urothelial carcinoma patients with an FGFR3 mutation, such as FGFR3 N540K, V553M, V555L/M, or L608V, and in 50% with FGFR4-mutant hepatocellular carcinoma, resistance has been observed [97]. Selective single-target FGFR inhibitors like *lirafugratinib*, LOXO-435, *roblitinib*, and *fisogatinib* have shown promise. *Bemarituzumab*, a monoclonal antibody, targets FGFR2b specifically, reducing the risk of adverse events associated with broad inhibition [98]. However, there is a potential risk of developing resistance by bypassing targets across isoforms.

### 3.3. Colony-Stimulating Factor 1 Receptor (CSF-1R)

CSF-1R is an RTK that is commonly found in monocytes, macrophages, and granulocytes. CSF-1R plays a role in promoting tumor immune evasion and therapy resistance by stimulating the proliferation, survival, and differentiation of tumor-associated macrophages [72,99]. Inhibiting CSF-1R signaling can alter the TME and boost immune responses. High levels of CSF-1R expression have been observed in various solid tumors like breast cancer, ovarian cancer, and endometrial tumors, correlating with poorer survival rates [100]. *Pexidartinib*, a Type III inhibitor, effectively targets CSF-1R and KIT with high potency. Its mechanism of action involves depleting immunosuppressive cells [43]. The efficacy of *pexidartinib* was demonstrated in the ENLIVEN trial (NCT02371369), a clinical trial involving 120 patients with tenosynovial giant-cell tumors. Patients treated with *pexidartinib* showed a 38% ORR, with 15% achieving a CR and 23% achieving a partial response (PR). In contrast, the patients in the placebo group had an ORR of 0%. Herein, the FDA approved *pexidartinib* for symptomatic *tenosynovial* giant-cell tumors not suitable for surgical improvement in 2019 [101].

CSF1R is unique compared to traditional SMKI targets as it primarily resides on tumor-associated macrophages (TAMs) within the tumor immune microenvironment. Recent studies have highlighted the significance of CSF-1R inhibition in reducing neuroinflammation, particularly in Alzheimer’s disease. Research has shown that CSF-1 signaling can predict the response of advanced NSCLC to ICIs. Preclinical studies have demonstrated that depleting TAMs using CSF-1R inhibitors can enhance the effectiveness of PD-1/PD-L1 blockades in various cancer models. Therefore, there was a hypothesis that *pexidartinib* could deplete TAMs in the TME, restore T-cell-mediated tumor clearance, and enhance the response to durvalumab. However, the outcomes did not meet expectations, as *pexidartinib* was found to inhibit FLT3-dependent dendritic cell (DC) differentiation, potentially counteracting the efficacy of durvalumab in patients with advanced cancer. This lack of efficacy in the combination treatment of *pexidartinib* and durvalumab may be attributed to the broad inhibition profile of *pexidartinib* [102]. Becker et al. discovered that the simultaneous inhibition of CSF1R and IL-6R could prevent cDC2 from transitioning into immunosuppressive tumor-induced DC3, thereby improving the therapeutic potential of DC-driven treatments [103]. CSF1R has demonstrated promise in targeting tumors and the TME. Targeting both IL-10 and CSF-1R with a bifunctional antibody has led to an expected reduction in tumor-associated macrophages and has also stimulated the proliferation, activation, and metabolic reprogramming of CD8+ T-cells, showing significant anti-tumor activity in various cancer models, particularly in head and neck cancer [104]. In addition, the multi-target drug *Sulfatinib* selectively targets FGFR, CSF-1, CSF-1R T663I mutations and VEGFR and has been approved by the National Medical Products Administration (NMPA) for the treatment of pancreatic and pancreatic neuroendocrine tumors. There are also dual CSF-1R/c-Kit inhibitors with improved stability and blood–brain barrier (BBB) permeability [99].

### 3.4. Cellular Mesenchymal-to-Epithelial Transition Factor (c-MET)

c-MET, alternatively named the hepatocyte growth factor receptor, plays a crucial role as the oncogene encoded by MET. c-MET exhibits high expression in tumors such as lung, esophageal, renal, and gastrointestinal cancers, playing a significant role in resistance to SMKIs and becoming an attractive therapeutic target [105].

*Capmatinib* and *tepotinib* are both highly potent Type I MET inhibitors that have been evaluated for treating NSCLC with MET exon 14 skipping mutations [32]. *Capmatinib*, with an IC50 of 0.13 nmol/L, was studied in the GEOMETRY trial with 97 patients [106], while *tepotinib* was assessed in the VISION trial with 152 patients. *Tepotinib* showed superior efficacy to *capmatinib* in previously treated patients, with an ORR of 43% and a median DOR of 10.8 and 11.1 months, respectively [107]. Both inhibitors have received FDA approval for metastatic NSCLC with MET exon 14 skipping alterations. While these mutations lead to decreased MET degradation, the efficacy of Type I MET inhibitors does not seem to be affected by these specific mutations [105]. However, mutations such as c-Met D1228 and Y1230 may confer resistance to these inhibitors. Although MET gene alterations are relatively low in frequency, they are significantly associated with resistance to other targets, especially EGFR inhibitors, by bypassing the MAPK pathway downstream of EGFR activation through MET status changes. MET amplification overexpression is associated with acquired resistance to the third-generation EGFR inhibitor *osimertinib* in advanced NSCLC patients with the EGFR T790M mutation [105]. This suggests that the dual targeting of the MET and EGFR could help overcome resistance in advanced NSCLC patients. In fact, the combination of *osimertinib* and the MET inhibitor *volitinib* achieved a response rate of over 65% in EGFR-mutant NSCLC patients who had previously received EGFR TKI treatment and progressed to MET amplification [108].

### 3.5. Proto-Oncogene Tyrosine–Protein Kinase Receptor (RET)

The RET gene encodes a transmembrane RTK, which plays a crucial role in various cellular processes. Various RET gene alterations have been identified as key drivers in the growth and proliferation of cancer cells. This highlights RET expression and function as potential therapeutic targets [109]. FDA-approved drugs like ponatinib, *cabozantinib*, sorafenib, and sunitinib can inhibit RET WT, serving as first-generation inhibitors and enabling the repurposing of other SMKIs [110]. *Pralsetinib* is a Type I inhibitor that potently inhibits both wild-type and activating RET mutants, including RET V804L, V804M, M918T, and C634W, as well as KIF5B-RET, CCDC6-RET, and CCDC6-RET (V804M) fusion [69]. The efficacy of *pralsetinib* was evaluated in ARROW(NCT03037385) in patients with RET fusion-positive metastatic NSCLC. A total of 87 patients were included, consisting of metastatic RET fusion-positive NSCLC patients who had progressed after platinum-based chemotherapy and treatment-naïve metastatic NSCLC patients. The results indicated an ORR of 57% [111]. Another Type I RET inhibitor, *selpercatinib*, potently inhibited RET WT, M918T, C634W, and CCDC6-RET and KIF5B–RET fusions. In LIBRETTO-001 (NCT03157128), a total of 105 patients with advanced or metastatic RET fusion-positive NSCLC were enrolled to evaluate the efficacy of *selpercatinib*. The trial results showed an ORR of 64% and a median DOR of 17.5 months, both higher than those observed in the ARROW trial with *pralsetinib* [33,112,113]. Therefore, both have been FDA-approved for patients with metastatic RET fusion-positive NSCLC.

In terms of resistance mechanisms, RET V804M/L in the gatekeeper region confers resistance to first-generation RET inhibitors. Additionally, MDM2 amplification has been identified as a potential mechanism of primary or acquired resistance to *cabozantinib* in NSCLC. Recent studies on resistance to second-generation RET inhibitors have revealed mutations at the RETG810R/S/C/V, Y806C/N, and βV738A sites within the RET kinase domain as drivers of acquired resistance to *pralsetinib* and *selpercatinib* [114]. Acquired KHDRBS1–NTRK3 fusion has been reported in patients with KIF5B–RET-fusion lung cancer after *selpercatinib* treatment, along with RET-altered cancers exhibiting NTRK and ALK fusions. This indicates a potential dual inhibition strategy targeting TRK and ALK [115]. RET fusion, similar to MET amplification, is a mechanism of acquired resistance to *osimertinib* in EGFR mutation-positive NSCLC. Combining *osimertinib* and *pralsetinib* may help alleviate resistance. Furthermore, MET gene amplification, through bypass activation, is implicated in off-target resistance in RET-rearranged NSCLC, potentially leading to resistance to *selpercatinib* and *pralsetinib*. These findings suggest a cross-interaction of RET mutation sites with other kinase mutations in RET-targeted therapy. The involvement of various targets, not only in tumor activation but also in cross-acquired resistance, underscores the importance of multi-target inhibition in overcoming resistance [116].

### 3.6. ROS Proto-Oncogene 1 (ROS1)

ROS1, a member of the RTK family, is a known oncogenic driver in various cancers, particularly lung adenocarcinoma. Despite being observed in only 1% to 2% of NSCLC patients [117], ROS1 plays a significant role in cancer through fusion proteins resulting from ROS1 rearrangements. These fusions, such as CD74–ROS1, SLC34A2–ROS1, and others, cause an uncontrolled activation of downstream pathways due to the loss of regulatory domains [118]. While FDA-approved multi-targeted SMKIs like *lorlatinib*, *entrectinib*, and *cabozantinib* have shown inhibitory effects on ROS1, the common G2032R mutation in ROS1 leads to resistance to first-generation inhibitors. *Repotrectinib*, a new-generation ROS1 inhibitor, has demonstrated efficacy against various ROS1 fusion-positive cancers and the G2032R mutation. The TRIDENT-1 trial (NCT03093116) evaluated repotrectinib in locally advanced or metastatic NSCLC patients with a positive ROS1 status, showing promising results in both inhibitor-naïve and pretreated patients, with effective responses observed against specific mutations [119]. *Repotrectinib* has been approved for use in adult patients with locally advanced or metastatic ROS1-positive non-small-cell lung cancer, although not all G2032R patients respond to this treatment.

ROS1 G2032R and MET amplifications are the most common on-target and off-target alterations associated with resistance to ROS1 inhibitors. Recent studies have identified the ROS1 F2004V mutation, which can still be inhibited by *repotrectinib* [120]. Zhao et al. identified through molecular dynamics simulations that patients with the ROS1 L2010M mutation may exhibit resistance to *lorlatinib*, *entrectinib*, *cabozantinib*, and *crizotinib*, while those with the ROS1 G1957A mutation may show resistance to *ceritinib* and *cabozantinib* [118]. However, there is still a relative lack of research on resistance to *repotrectinib*, and further exploration of its resistance mechanisms is needed. A study reported resistance mechanisms in NSCLC with MET amplification and ROS1 rearrangement treated with the ROS1/MET dual inhibition strategy of *lorlatinib* plus *crizotinib*, where the subsequent emergence of MET-targeting resistant mutations and a loss of MET amplification highlighted tumor cell resistance to sequential targeted therapy [121].

### 3.7. Protein Kinase B (AKT)

The excessive activation of the PI3K/Akt/mTOR signaling pathway is commonly observed in solid tumors. Mutations in genes upstream of this pathway or in AKT itself can lead to a hyperactivation of AKT, comprising AKT1, AKT2, and AKT3, resulting in the phosphorylation of various substrates. This alters substrate activity, inhibiting apoptosis, promoting cell survival and proliferation, and reprogramming cell metabolism [122,123]. *Capivasertib* is a Type I inhibitor of the serine/threonine kinase AKT, targeting all three isoforms and inhibiting the phosphorylation of downstream AKT substrates. A trial (Capello-291, NCT04305496) assessed the efficacy of *capivasertib* in combination with fulvestrant in patients with inoperable or metastatic HR-positive and HER2-negative breast cancer, including those with PIK3CA/AKT1/PTEN alterations. The results showed an ORR of 26% in the *capivasertib* with fulvestrant group compared to 8% in the placebo with fulvestrant group, with a median DOR and median progression-free survival (PFS) of 10.2 vs. 8.6 months and 7.3 vs. 3.1 months, respectively. Consequently, *capivasertib* has become the first FDA-approved AKT inhibitor for use in HR-positive and HER2-negative breast cancer patients with one or more PIK3CA/AKT1/PTEN alterations [31].

Inhibiting the PAM pathway has been a hot topic in cancer drug development for decades, leading to the development of more than 70 inhibitors targeting PAM [122]. However, only a very few have been approved for cancer indications. Due to the multiple upstream signals activating the PAM pathway and its crosstalk with various pathways, which are widely involved in normal physiological activities and tumorigenesis, the most common issue with PAM pathway inhibitors is their toxicity and side effects [124]. Several inhibitors targeting PI3K have received FDA approval, yet concerns remain regarding resistance, sensitivity markers, and toxicology, possibly due to the broad suppression of PI3K subtypes. For instance, the PI3Kδ and PI3Kγ subtypes are closely related to immune regulation; inhibiting PI3Kδ can suppress both innate and adaptive immune systems, making patients susceptible to severe infections. Furthermore, autoimmune pneumonitis and colitis are often associated with PI3Kδ inhibition [125]. The mTOR inhibitor sirolimus was approved in 1999 for immunosuppression in kidney transplantation, yet its application in cancer still poses significant challenges. The only mTOR inhibitor approved for cancer treatment is *temsirolimus*, which was approved in 2007 for renal cell carcinoma patients, and there have been no mTOR inhibitors approved since then. Currently, there is still a lack of research on AKT inhibitor resistance, which requires further investigation. The AKTE17K mutation has been reported to be correlated with the response to the AKT inhibitor *capivasertib* [125].

## 4. New Strategies for Kinase-Targeted Cancer Therapies

### 4.1. Monoclonal Antibodies and Derivatives

RTKs in kinases are excellent targets for mAbs. However, due to the large molecular weight of intact mAbs, mAbs targeting RTKs have difficulty penetrating into the interior of tumors. Therefore, a series of derivatives have been produced through antibody modification to enhance penetration and cytotoxicity. Among these, developing smaller nanobodies enhances antibody penetration, reduces immunogenicity, and increases efficacy while decreasing side effects.

#### 4.1.1. Monoclonal Antibodies

We have compiled a list of mAbs targeting RTKs, excluding biosimilars, currently in Phase III clinical trials (Table 2). Here, we mainly focus on mAb drugs in Phase III clinical trials that are not yet approved. Insulin-like growth factor 1 receptor (IGF-1R) contains transmembrane and tyrosine kinase domains, belonging to the RTK family. The activation of IGF-1R stimulates the PI3K/AKT signaling pathways, leading to the generation of pro-survival and proliferative signals [126]. Currently, the abti-IGF-1R mAbs ganitumab and VRDN-001 are in Phase III clinical trials. A double-blind Phase III study evaluated the efficacy and safety of ganitumab in combination with gemcitabine for metastatic pancreatic cancer. This combination exhibited manageable toxicity but did not improve overall survival [127]. Another Phase III trial of ganitumab in metastatic Ewing sarcoma patients showed that it did not significantly reduce the risk of event-free survival (EFS) and may be associated with increased toxicity [128]. Clinical trials of ganitumab in bone metastases and secondary lung malignancies are ongoing. Receptor tyrosine kinase-like orphan receptor 1 (ROR1), a member of the RTK family, is a tumor embryonic antigen expressed in various tumors, including chronic lymphocytic leukemia (CLL), ovarian cancer, and endometrial cancer [129]. Zilovertamab is a humanized monoclonal antibody targeting ROR1 that is capable of blocking Wnt5a-induced ROR1 signaling. It is currently being assessed in Phase III trials for T-cell lymphoma [130]. Additionally, the Phase II trial results of the selective targeting of FGFR2b with bemarituzumab indicated that although there was no statistically significant improvement in PFS, the combination of bemarituzumab and -mFOLFOX6 resulted in an elevated median PFS and OS compared to monotherapy. Phase III trials are also currently investigating this combination in advanced gastric or gastroesophageal junction adenocarcinoma patients overexpressing FGFR2b (NCT05052801, NCT05112626) [98,131].

#### 4.1.2. Nanobodies

Due to their relatively large size and physicochemical properties, mAbs exhibit an uneven distribution in the TME, are typically restricted to the perivascular space, and face challenges in penetrating the blood–brain barrier. Nanobodies, derived from the antigen recognition domain of camelid heavy-chain antibodies, are ten times smaller than traditional antibodies and have a molecular mass of about 15 kDa (Figure 3a). This smaller size enhances tumor penetration, allowing nanobodies to reach cells within poorly perfused tumor regions [132]. Puttemans et al. compared the therapeutic effects of anti-HER2 nanobodies and trastuzumab in an intracranial human ovarian cancer brain metastasis nude mouse model established through intracerebral injection. Impressively, this brain metastasis model allowed the anti-HER2 nanobody to penetrate the BBB, whereas trastuzumab could not, resulting in only the 131I- or 225Ac-labeled nanobody extending the survival time, while trastuzumab alone was ineffective [133]. Nanobodies are rapidly cleared from the circulation, resulting in a quick target-to-background contrast and making them highly suitable for molecular imaging but possibly less ideal for therapy. Currently, the vast majority of nanobodies are still in the preclinical stage; however, radiolabeled nanobodies have entered the clinical stage. [134]. 131I-GMIB-anti-HER2-VHH1 is a nanobody-based targeted radionuclide theragnostic agent. Phase I trials have demonstrated its favorable toxicity profile and uptake in HER2-positive lesions, making it potentially applicable for patients after trastuzumab treatment; Phase II clinical trials of this agent are underway [135]. In contrast, the diagnostic radiopharmaceutical Ga68-RAD207 based on the PTK7 nanobody for PET imaging ceased development in the preclinical stage.

Being bound to albumin significantly increases the half-life of nanobodies, which is advantageous for developing cancer therapeutic agents. Recent studies have shown that nanobodies can overcome the resistance of mAbs. Liu et al. developed a nanobody targeting the HER2-ECD domain I, which, under trastuzumab-resistant conditions, exhibited significantly enhanced synergistic antitumor effects compared to pertuzumab [136]. However, nanobodies retain only the targeting fragments and lack the Fc portion responsible for activating the complement system and cytotoxicity pathway, thereby reducing the antibody’s tumor-killing effect [17]. Therefore, nanobodies are typically combined with therapeutic molecules and other drug carriers (such as cytotoxic drugs and immunotoxins). Nanobody-based multivalent nanobodies, bispecific antibodies, and nanobody–drug conjugates have also been developed. The main advantage of the lack of an Fc segment and the highly homologous sequence with human VH domains is the reduction in side effects caused by immune reactions [17,137,138].

### 4.2. Kinase Degraders

SMKIs and mAb-based developments both require specific, accessible binding sites or conformations on the target, whether it be the ATP-binding domain, allosteric sites, or the extracellular domain of RTKs. However, some kinases lack universal conformations or changes due to resistance mutations or truncations in the extracellular domain of RTKs, such as p95 HER2, EGFRvIII, and FGFR2 ΔE18 [139]. This leads to challenges in developing SMKIs or mAbs. Recent studies have found that the BTK L528W mutation can activate downstream BCR signaling in a kinase activity-independent manner, resulting in resistance to covalent and non-covalent BTK inhibitors [140]. Additionally, in the presence of oncogenic RAS, the inhibition of BRAF can drive the formation of BRAF–CRAF dimers, activating downstream signaling pathways and leading to tumorigenesis [141]. These mechanisms can render SMKIs or mAb-based strategies ineffective. Therefore, employing targeted protein degradation strategies to selectively degrade kinases can expand kinase inhibition methods. Targets previously deemed undruggable such as p53 and RAS may be developed into drugs through targeted protein degradation (TPD) strategies [142,143]. TPD mainly consists of two categories: proteolysis-targeting chimeras (PROTACs) and molecular glues (Figure 3b). We summarize the clinically tested PROTACs and molecular glue degraders targeting RTKs in Table 3. The latest protein degradation strategies also include lysosome-targeting chimeras (LYTACs), such as endosome–lysosome and autophagy–lysosome, and new degradation methods like dTAG, hydrophobic tagging, and Trim-Away [144,145,146].

#### 4.2.1. PROTACs

PROTACs are bifunctional molecules that mediate the proximity of E3 ligases and the protein of interest (POI) by linking them via ligands (see Figure 3b), thus leading to the ubiquitination and proteasomal degradation of the POI [148] and enabling the modulation or inhibition of its function. The advantage of PROTACs lies in their high selectivity for target proteins through the binding of target ligands, facilitating efficient degradation; therefore, they have the potential to be designed against various proteins.

Currently, most research on PROTACs is in the preclinical stage, with only a few PROTACs targeting kinases entering clinical trials, primarily focusing on BTK and EGFR targets. CFT8919 is an orally bioavailable targeted EGFR L858R protein degrader used to treat NSCLC that has acquired an FDA Investigational New Drug (IND) status [149]. In preclinical studies, CFT8919 has shown promise in overcoming resistant mutations such as EGFRT790M, L858R, and C797S, which cause acquired resistance to first-, second-, and third-generation EGFR inhibitors [150]. UBX-303-1 is an oral degrader targeting BTKs that is used for treating B-cell malignancies and has received FDA IND approval. In preclinical studies, UBX-303-1 has demonstrated enhanced target selectivity and potent tumor growth inhibitory activity, effectively degrading various resistant mutations of the BTK protein [151]. Another PROTAC, BGB-16673, which targets wild-type BTK and multiple resistant mutations, showed promising results in a Phase 1 study (NCT05006716) for patients with B-cell malignancies, with an ORR of 57% (n = 28) [152]. CFT1946 selectively inhibits and degrades the mutant BRAF V600 protein while sparing WT BRAF V600, demonstrating preclinical activity in BRAF V600 mutant models (including BRAFi-resistant models) [153]. These findings indicate promising potential for PROTACs across various point mutations, yet the results of the clinical trial are still pending.

PROTACs typically have higher molecular weights, resulting in lower oral bioavailability and difficulty in penetrating cell membranes, often leading to clinicians using intravenous administration and causing off-target effects in normal tissues and severe side effects [148]. Therefore, one of the directions in PROTAC drug development is to improve bioavailability, which is particularly crucial for orally administered compounds. The E3 ligase ligand and linker substantially influence the water solubility of PROTACs. Therefore, the PROTAC linker region has garnered significant attention. Although there are no universally accepted linker design principles to ensure target protein degradation, enhanced solubility and permeability have been demonstrated [154]. Cantrill et al. achieved satisfactory solubility and permeability by altering the linker and E3 ligand by removing polar carbonyl groups [155].

#### 4.2.2. Molecular Glue Degraders

Small-molecule protein degraders, known as molecular glue degraders, have been developed and utilized to improve bioavailability. Molecular glue is a monovalent molecule that binds to the surface of E3 ligase receptors, achieving binding to the target protein through protein–protein interactions and leading to the degradation of the POI (Figure 3b) [156]. Compared with PROTACs, molecular glues have advantages such as ease of cell membrane penetration, excellent metabolic properties, and suitability for oral administration. The immunomodulatory inhibitor thalidomide (with next-generation derivatives lenalidomide and pomalidomide) was unexpectedly discovered as a molecular glue drug; however, its severe teratogenic effects were discovered in the 1960s [157]. Thalidomide binds to cereblon (CRBN), remodeling its surface domain to capture undiscovered proteins and induce proteasomal degradation. It has been FDA-approved for malignancies such as multiple myeloma and myelodysplastic syndrome. However, the exact target proteins of immunomodulatory inhibitors remain incompletely elucidated. Lenalidomide has been shown to trigger CK1α degradation via ubiquitination through the CRBN–CRL complex. CK1α, a multifunctional serine/threonine kinase, acts as a negative regulator of the β-catenin and p53 pathways, and its complete loss activates p53 and apoptosis in hematopoietic stem cells [158].

Molecular glue degraders targeting kinases are still in the preclinical stage of development. The dysregulation of the serine/threonine kinase NIMA-related kinase 7 (NEK7) leads to abnormal IL-1β production and is involved in the activation of NLRP3 inflammasomes [159]. In the context of cancer, high levels of IL-1β in the TME exert potent immunosuppressive effects. CT01, a small-molecule glue targeting NEK7 developed by Captor Therapeutics, can induce the degradation of proteins including NEK7, GSPT1, and SALL4 and has been applied to the treatment of hepatocellular carcinoma, lung cancer, and NET tumors [160]. NK-7-902, a novel molecular glue degrader targeting NEK7 developed by Novartis, has not yet been fully evaluated in the literature.

Previously reported molecular glue degraders reshape the surface domain of E3 ligases to capture the POI. However, CR8 binds to the active site of the target protein CDK12 and is recruited to the CUL4–RBX1–DDB1 core complex through DDB1, driving the CR8-induced degradation of cycK [161]. This indicates that small molecules binding to target proteins can recruit E3 ligases to induce the degradation of POIs, thereby expanding the development pathways of molecular glue degraders. Ma et al. conducted molecular docking analysis screening and found that the small-molecule compound A80.2HCl is associated with CDK4, CDK6, and MYC degradation. A80.2HCl interacts with CRBN and MYC, inducing MYC degradation at nanomolar concentrations, re-establishing the sensitivity of MYC-overexpressing cancer cells to CDK4/6 inhibitors [162] and indicating the potential role of virtual screening for small-molecule degraders. The use of molecular glues is no longer limited to the interaction between E3 ligases and POIs but has expanded to protein–protein interactions (PPIs). This is theoretically applicable to all protein interaction processes, as will be discussed below.

#### 4.2.3. Other Degraders

In addition to PROTACs and molecular glues, several new TPD strategies have shown promise. LYTACs are hybrid molecules that simultaneously bind to the cell surface the lysosome-targeting receptor (LTR) and extracellular proteins, inducing the internalization of target proteins and their degradation via the lysosomal pathway. Banik et al. successfully induced the degradation of proteins including EGFR, APOE4, CD71, and PDL1 by conjugating small molecules or antibodies targeting the extracellular domains of the target proteins with the LTR family member CI-M6PR [163]. This provides a direction for degrading secreted and membrane proteins via lysosomal degradation, offering broad application prospects. However, due to the retrotranslocation complex leading to recycling from the endosome to the plasma membrane, the first generation of LYTACs has been unable to achieve more than 70 to 80% target degradation. Recent research suggests that engineering LYTACs with antibodies can promote the release of the target protein in low pH conditions, significantly enhancing degradation efficiency without relying on LYTAC recycling [164]. Recently, antibody-based PROTACs (AbTACs) have been developed under receptor-specific degradation, in which molecules targeting E3 ligases are conjugated with antibodies. Unlike antibody–drug conjugates, AbTACs release the active PROTACs upon intracellular entry through linker hydrolysis, subsequently inducing the degradation of the target protein. Richa et al. coupled pomalidomide with cetuximab to degrade EGFR receptors on cancer cells, aiming to provide a broader therapeutic window while reducing the risk of off-target toxicity [165].

AUTACs are dual-function molecules with a ligand connecting the POI and an S-guanosylation autophagy-recruiting tag, triggering the K63 polyubiquitination of the POI. This polyubiquitination allows the POI to be recognized by SQSTM1/p62 for degradation through selective autophagy pathways [166]. Yong et al. developed an Autophagy-targeting chimera (AUTOTAC), which does not rely on polyubiquitination. Instead, it directly interacts with the ZZ domain of SQSTM1/p62 (p62zz), targeting the POI for autophagic degradation [167]. Trim-Away technology employs commercially accessible antibodies alongside TRIM21 to swiftly degrade proteins. TRIMbody, developed by Chen et al., fuses the POI with a nanobody and the RBCC domain of TRIM21. This system overcomes the difficulty that antibodies have in penetrating the cell nucleus and membrane. Hydrophobic tagging-based protein degradation (HyT-PD) involves hydrophobic tags, linkers, and POI ligands. It binds hydrophobic fragments to the target protein’s surface, mimicking misfolded or damaged proteins and leading to their degradation [168]. Also, degradation tags (dTAGs) rely on specific labels for targeted protein degradation [145]. However, these emerging protein degradation technologies are still lacking in terms of their development as drugs targeting kinases. It can be anticipated that these methods will expand the strategies for kinase degradation.

TPD shows strong potential for novel drug development and overcoming resistance to SMKIs and mAbs. The newly developed PROTACs can target multiple point mutations, such as CFT8919 and UBX-303-1. However, since TPD relies on specific E3 ligases, tumor cells may downregulate these ligases, leading to resistance. The downregulation of the key E3 ligase CRBN has been observed in resistance to immunomodulatory inhibitors and PROTACs in preclinical studies [169]. Therefore, there is a need to develop new E3 ligases. Efforts are underway to develop DCAF15, DCAF16, IAPs, RNF114, and VHL ligands to expand the range of E3 ligases. Recently, strategies for extracellular targeted protein degradation (eTPD) have emerged. Theoretically, intracellular and extracellular target proteins can be degraded through TPD, providing a new direction for developing degraders targeting RTKs [170]. It is worth noting that TPD leads to the complete degradation of the target protein. Once adverse reactions are triggered by the degradation of the target protein, cells may have difficulty restoring target protein expression in a short timeframe, thus the duration of side effects may be prolonged.

### 4.3. Protein–Kinase Interaction Inhibitors (PKIIs)

The biological activity of kinases is regulated through protein complexes and typically mediated by PPIs. In recent years, PPIs have emerged as potential therapeutic targets. Protein–kinase interaction inhibitors (PKIIs) inhibit the kinase and its substrate PPIs; this differs from the mechanism of action of SMKIs, which relies on ATP-binding pockets or allosteric sites. However, the larger interface areas involved in PPIs, ranging from 1500 to 3000 Å2, compared to the smaller receptor–ligand contact area (300–1000 Å2), present a highly hydrophobic interface devoid of grooves or pockets and pose challenges for the binding of small-molecule drugs [171]. It should be noted that some allosteric inhibitors are also believed to act on PPI sites, but they inhibit kinase activity, while PKIIs inhibit the interaction between kinases and substrates [172]. Unlike small molecules, peptides exhibit more flexible conformations, higher selectivity, and lower costs than antibodies, making them ideal candidates for inhibiting PPIs [19], as shown in Figure 3c. Short linear peptides, in particular, have advantages in improving drug properties and limiting toxicity. Currently, the development of PKIIs is still in the preclinical stage. VAL201 is a decapeptide designed to inhibit the interaction of Src with the androgen or estradiol receptors to inhibit growth without blocking desirable receptor-dependent transcriptional activity. This approach aims to eliminate the majority of side effects associated with androgen therapies [173]. However, two Phase I clinical trials (EUCTR2013-004009-25-GB, NCT02280317) did not show satisfactory results with VAL201 [174,175]. Through computer analysis, Mallick et al. identified the key TrkA-binding peptide sequence of NGF and designed a hybrid peptide candidate that blocks the interaction between NGF and TrkA, which showed potential in the treatment of ameloblastoma [176]. Mulate et al. used AlphaFold to identify various peptides targeting PKC [19]. It is expected that more peptide-based compounds will enter development and clinical research as PKIIs.

The concept of molecular glues has expanded beyond the interaction between a POI and an E3 ligase. In addition to molecular glue degraders, some molecular glues induce protein–kinase binding through PPIs, leading to kinase inhibition by promoting an inactive conformation. These are classified as non-degrading molecular glues. IK-595, for example, can form inactive complexes between MEK and all RAF subtypes and BRAF mutants, thereby blocking RAF-dependent MEK phosphorylation and attenuating CRAF-mediated MEK reactivation [177]. Haines et al. demonstrated that IK-595, compared to other MEK/RAF inhibitors, can inhibit MEK and ERK1/2 phosphorylation for a longer duration in RAS-mutant cancer models, achieving prolonged target engagement and pathway inhibition. Moreover, IK-595 has enhanced the efficacy of conventional chemotherapy drugs in KRAS-driven tumor models, broadening the potential clinical applications of IK-595. A Phase I clinical trial (NCT06270082) of IK-595 is currently underway [177]. Another molecular glue targeting RAF–MEK, known as NST-628, was developed to fully abrogate MEK, ERK, and RAS–MAPK phosphorylation by stabilizing all RAF–MEK complexes, including CRAF–MEK, BRAF–MEK, and ARAF–MEK. Compared to clinically approved RAS–MAPK pathway inhibitors, NST-628 robustly and persistently reduces pathway reactivation in KRAS–G13D tumor models. Additionally, NST-628 completely blocks RAF paralog dimerization in RAS-driven cells, which may help reduce pathway reactivation in biomarker-driven tumors. A Phase I clinical trial (NCT06326411) is currently ongoing [178]. Additionally, despite small molecules being considered challenging for PKII development, there have been recent reports of small-molecule PKIIs. Gao et al. synthesized a series of CDK9–cyclin T1 PPI inhibitors based on 4,4′-bipyridine derivatives. The CDK9–cyclin T1 complex phosphorylates RNA polymerase, promoting pro-tumorigenic factor expression and transcriptional activity (C-myc and Mcl-1). Among these, B19 exhibited higher inhibitory effects on breast cancer cell lines than ATP enzyme inhibitors and does not rely on ATP inhibition but directly disrupts CDK9–cyclin T1 interactions. The anticancer effect of B19 was validated in a xenograft mouse model, and it showed synergistic effects with olaparib [179].

In addition to directly inhibiting protein kinases, disrupting kinase stability can also be a strategy to prevent signal transduction. The Hsp70–Bim dimer is a specific target activated by BCR–ABL in chronic myeloid leukemia that can lead to TKI resistance by stabilizing downstream kinases such as AKT and Raf-1. S1g-2, an inhibitor of the Hsp70–Bim interaction developed by Song et al., has shown antitumor effects in vitro and in vivo by disrupting AKT, Raf-1, and other kinases, especially in TKI-resistant CML [180].

However, current PPI inhibitors focus more on “undruggable” targets such as Bcl-2, MDM2-p53, and SMAC-XIAP [181]. Research on PKIIs in the clinical stages of development is still limited, possibly due to the established system for developing SMKIs and the non-conservative PPI conformations. However, with a deeper understanding of protein structures and the rapid progress of molecular docking and AI (such as AlphaFold), more accurate predictions of inhibitor–kinase interactions will accelerate the development of PKIIs.

## 5. Future Perspectives

As of 1 March 2024, more than 120 SMKIs have received global approval for treating various diseases, with almost 70 of them approved by the FDA specifically for cancer treatment [28]. This review provides a comprehensive overview of the mechanisms and targets of recently FDA-approved SMKIs. It also delves into emerging kinase inhibition strategies, including monoclonal antibodies and their derivatives, kinase degraders, and protein kinase interaction inhibitors. Despite these advancements, challenges persist in the development of kinase inhibitors. Only a fraction of the entire kinase family has been explored as potential targets, with 70% of the well-researched TK family’s targets yet to be developed and 30% of the kinase family remaining inadequately studied [182]. This highlights the substantial opportunity for further advancement in kinase inhibition strategies [5], such as identifying new targets, addressing drug resistance, and overcoming challenges like crossing the blood–brain barrier. Computational techniques and combination therapies offer promising avenues for enhancing the development of kinase inhibitors.

### 5.1. Potential New Targets

Currently, several kinase inhibitors targeting novel potential targets have progressed to advanced clinical stages and show promise for future approval (Table 4). In this summary, we focus on Aurora kinases (AURKs), which are serine/threonine kinases involved in centrosome maturation, spindle assembly, chromosome alignment, and cytokinesis during mitosis (AURK-A, B) and meiosis (AURK-C) [183]. The overexpression of AURKs is observed in various cancers and is linked to metastatic tumor invasiveness and resistance to chemotherapy [184]. AL-8326, a multi-target kinase inhibitor that inhibits Aurora B, FGFRs, and VEGFRs, has shown clinical benefits in Phase Ib/IIa studies for monotherapy in ≥ third-line SCLC patients, with ongoing Phase II and III trials [185]. Additionally, *Tinengotinib* (TT-00420), another multi-kinase inhibitor targeting Aurora A/B, has progressed to Phase II/III clinical trials. A Phase I trial for advanced solid tumors (NCT03654547) demonstrated a partial response in 30.2% of patients [186]. The most recent Phase Ib/II results reported an ORR of 15.8% in 19 patients receiving *tinengotinib* monotherapy [187].

WEE1 inhibits CDK through Y15 phosphorylation, thereby arresting the cell cycle at the G2-M and G1-S checkpoints to allow for DNA repair and prevent apoptosis. It is significantly upregulated in various TP53-mutant cancer types, including ovarian cancer, glioblastoma, leukemia, and breast cancer [188]. *Adavosertib*, a selective oral WEE1 kinase inhibitor, demonstrated efficacy in a Phase II study with recurrent uterine serous carcinoma patients, showing a 29.4% ORR [189]. Another Phase II trial with recurrent ovarian cancer patients indicated that *adavosertib* in combination with gemcitabine led to a prolonged median PFS [190]. The cell cycle regulatory kinases CHK1 and CHK2 are key targets of the ATM/ATR kinases and play a role in DDR pathways [191]. CHK1/2 inhibitors are currently in preclinical development. *Prexasertib*, a CHK1 and CHK2 inhibitor, has demonstrated clinical activity and tolerability in ovarian cancer patients [192]. Additionally, the CHK1 inhibitors LY-2880070 and CCT-245737 are currently undergoing Phase II clinical trials [193].

The Janus kinase (JAK) family, consisting of JAK1, JAK2, JAK3, and TYK2, is implicated in cancer and immune-related diseases. The aberrant activation of the JAK pathway can drive tumorigenesis by enhancing cell proliferation, invasion, and metastasis [194]. While JAK SMKIs have been approved by the FDA for autoimmune diseases, targeting the JAK family in oncology remains a challenge [195]. A Phase I study revealed limited clinical activity in late-stage solid tumor patients treated with a combination of pembrolizumab and *itacitinib* (a JAK1 inhibitor) [196]. Currently, *golidocitinib* (a JAK1 inhibitor) is under FDA review for peripheral T-cell lymphoma treatment. Phase II trials have demonstrated that *gandotinib* (a JAK2 small-molecule inhibitor) is well tolerated and associated with clinical improvements in patients with myeloproliferative neoplasms [197]. Promising results have also been observed with the PLK1 inhibitor *onvansertib* and the LYN inhibitor Masitinib mesylate.

### 5.2. Computing to Accelerate Drug Development and Repurposing

Artificial intelligence and machine learning have been extensively utilized in molecular design, computer screening, predicting protein–protein interactions, and drug repurposing. Computer screening and prediction can significantly expedite the development of new kinase inhibitors. Recent research has demonstrated the identification of new targets and drug-resistant mutations through genomics and computational methods. For instance, Iyer et al. leveraged public databases to screen ROS1 variants, employing the “Sorting Intolerant From Tolerant” algorithm to assess the impact of mutations on protein function, particularly focusing on those with deleterious effects. They further evaluated the influence of mutations on ROS1 inhibitors through in vitro experiments [117]. Similarly, Zhao et al. investigated the variations in binding affinity between the ROS1 protein and specific mutations like L2010M, G1957A, D1988N, and L1982V using molecular docking and molecular dynamics simulation techniques. Their MD simulation results indicated that patients with the ROS1 L2010M mutation might develop resistance to *lorlatinib*, *entrectinib*, *cabozantinib*, and *crizotinib*, while those with the ROS1 G1957A mutation could exhibit resistance to *ceritinib* and *cabozantinib* [118]. This methodology enables the early identification of on-target resistance mutations related to specific targets before drug development, facilitating preliminary assessments of drug efficacy against resistant variants.

Machine learning methods show promise in the repurposing of existing kinase inhibitors or compounds already in clinical use for cancer treatment and other diseases. Enrique et al. used computational techniques to identify mitoxantrone and abacavir as potential drugs for ALK-positive NSCLC and validated this combination in in vitro experiments [198]. Muthuraj et al. employed the t-SNE algorithm and ECFP4 fingerprints to screen derivatives of palbociclib and *ribociclib*, identifying DE50607359 as a promising WEE1 inhibitor, pending experimental validation [199]. Vishwakarma et al. utilized computational techniques to explore the inhibitory effects of FDA-approved drugs on PI3K, discovering lapatinib as a pan-Class I PI3K inhibitor and dual dapagliflozin as a γ-subtype-specific PI3K inhibitor [200]. While drug repurposing may increase side effect risks, it can expedite target point development based on existing structures. The approval of the first ALK inhibitor shortly after the discovery of oncogenic ALK alterations in lung cancer suggests that computational techniques can accelerate drug discovery through repurposing [201].

### 5.3. Overcoming Drug Resistance

In the pursuit of overcoming resistance to existing kinase-targeted drugs, researchers are exploring alternative and novel targets and developing new kinase-targeting strategies [6]. Previous discussions have focused on the emerging resistance mechanisms to new targets and the corresponding strategies to overcome them. The most comprehensive area of research on resistance mechanisms is related to EGFR inhibitors. Specific mutations such as EGFR L858R/Del19, T790M, and C797 have been identified as drivers of resistance to first-, second-, and third-generation EGFR inhibitors, respectively [202]. Various inhibitors are being developed to address the on-target mutations causing resistance to *osimertinib*, including allosteric inhibitors like JBJ-04-125-02 and EAI001, as well as ATP-competitive inhibitors such as BLU-945, BLU-525, BBT-176, and OBX02-011, which spare the wild-type EGFR [20]. Additionally, the bypass activation of pathways like MET, MEK, AXL, and FGFR can also lead to *osimertinib* resistance. Combining *osimertinib* with inhibitors targeting these bypass activation pathways could be a potential strategy, for example, using *osimertinib* in combination with *carbozantinib* (MET) or trametinib (MEK) [202]. *Brigatinib* is considered effective in overcoming the acquired resistance induced by AXL activation [203]. Kinase degraders are seen as another promising strategy, as they can inhibit kinase activity for extended periods through direct protein degradation, potentially reducing the likelihood of resistance. While research in this area is still limited, PROTACs in clinical trials show potential for overcoming resistance to small-molecule kinase inhibitors by targeting multiple point mutations. Additionally, PKIIs, with their broader contact interface and distinct mechanism from SMKIs, also show promise in overcoming resistance. These emerging strategies are still in the early stages and require further investigation to surpass the efficacy of existing treatment strategies.

### 5.4. Crossing the Blood–Brain Barrier

With the advancement of comprehensive cancer therapies, the survival rates of patients with advanced cancer have notably increased. Currently, most EGFR SMKIs are effective in treating NSCLC, where brain metastases significantly impact patient prognosis. However, crossing the BBB is crucial for the systemic treatment of tumors using kinase inhibitors. The third-generation EGFR inhibitor *osimertinib*, the ALK inhibitor *alectinib*, the RET inhibitor *selpercatinib*, and ROS1 or NTRK inhibitors like *entrectinib* have shown a central nervous system (CNS) ORR of up to 80%. However, other targets, such as EGFR exon 20 insertions, C797S mutations, HER2 mutations, and MET exon 14 skipping, still require the development of CNS-active small-molecule inhibitors [204]. A promising strategy in this field involves combining next-generation ROS1 TKIs with selective MET TKIs like *crizotinib* or *tepotinib*, which have the ability to penetrate into the CNS. Preclinical evaluation of drug concentrations in the CNS is essential for predicting BBB permeability. Pharmacokinetic models have been established for this purpose [205]. The unbound partition coefficient (Kp_uu_), comprising Kp_uu,brain_ and Kp_uu,CSF_, indicates the ratio of the free drug concentration in the brain tissue and cerebrospinal fluid to the free concentration of the same drug in the plasma at equilibrium. This measure evaluates CNS penetration and drug efflux by transport proteins at the BBB. From a clinical standpoint, objective metrics such as the CNS ORR and median PFS should also be considered when evaluating the CNS activity of drugs [204]. These objective measures can offer valuable insights into the efficacy of kinase inhibitors in managing CNS metastases.

### 5.5. Combination with Immunotherapy

In addition to kinase inhibitors, immunotherapy has shown significant potential in recent years, greatly improving the outlook for patients with lung cancer and other tumors. Exploring combinations using kinase inhibitors and immunotherapeutic agents can maximize the use of the current limited treatment options to provide greater clinical benefits to patients. Although combinations of ICIs with SMKIs have been approved for tumors like hepatocellular carcinoma, renal cell carcinoma, and endometrial cancer [206], they are associated with a high incidence of grade 3 or higher toxicity, reaching up to 82% [207], which poses challenges for clinical treatment approaches. Some studies propose that administering the ICI and the TKI sequentially instead of simultaneously could potentially overcome their severe toxicities without compromising their effectiveness [208]. Furthermore, kinases play a regulatory role in almost all immune cells within the TME. Various potential targets are emerging in the TME, including FAK receptors on fibroblasts, JAK, which is involved in pro-invasive phenotypes, c-KIT on MDSCs, LCK in T-cell receptor signal transduction, Tie2 receptors on monocytes/macrophages, and HCK, which influences the expression of immunosuppressive genes [209]. *Defactinib*, a FAK-targeting therapy, has been recognized by the FDA as a breakthrough treatment. A Phase I study demonstrated that combining *defactinib*, pembrolizumab, and gemcitabine in patients with advanced refractory infiltrating pancreatic cancer can activate T lymphocytes, indicating biomarker activity [210]. Developing kinase inhibitors with high selectivity that target the TME may enhance the efficacy of immunotherapy while minimizing toxic side effects.

In conclusion, kinases remain ideal targets for cancer therapy. It can be anticipated that various kinase-targeted strategies, including SMKIs, will continue to hold immense potential in future cancer therapy.

## Figures and Tables

**Figure 1 ijms-25-05489-f001:**
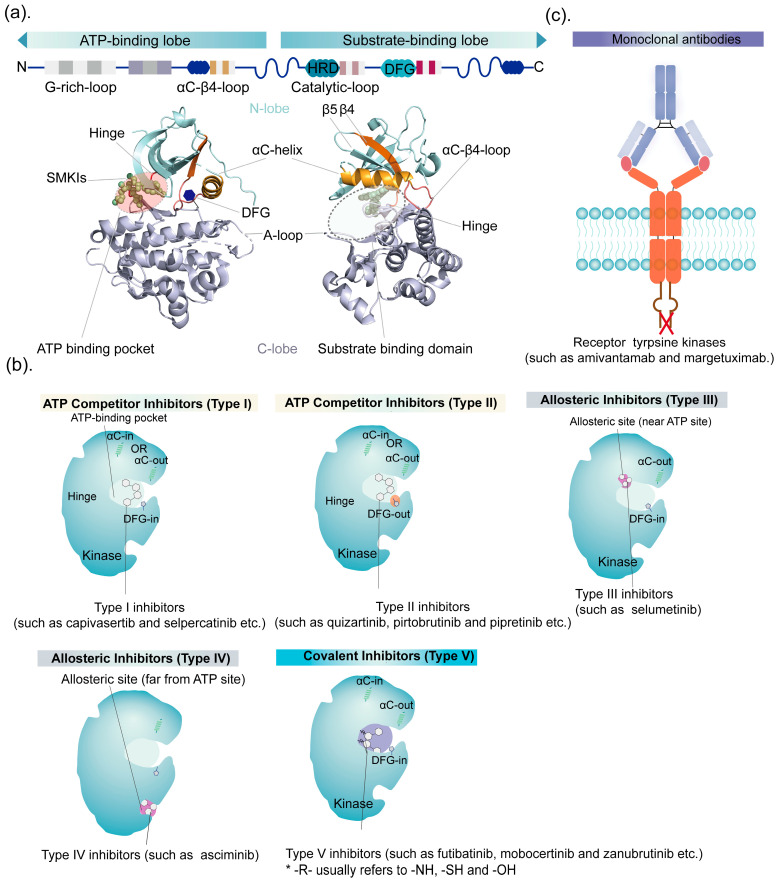
(**a**) The 2D and 3D general structures of kinases. Kinases consist of two parts, the ATP-binding lobe and the substrate-binding lobe. The G-rich loop plays a role in catalyzing phosphoryl transfer reactions within the catalytic core of protein kinases. The αC–β4 loop moves outward when the kinase is inactive and inward when the kinase is active. HRD is directly involved in autophosphorylation, controlling the activation state of protein kinases. DFG can chelate magnesium ions, and its movement is necessary for the active conformation. A-loop refers to the activation loop, with its N-terminus determined by the DFG motif. The 3D model was generated using PyMOL (Protein Data Bank entry: 4ZAU). (**b**) Mechanism of action of small-molecule kinase inhibitors. ATP-competitive inhibitors are classified as Type I or Type II. Type I kinase inhibitors competitively inhibit ATP by binding to amino acid residues in the ATP-binding pocket or hinge region, characterized by their ability to adopt the DFG-in conformation. Type II inhibitors bind to the inactive DFG-out conformation of kinases, regardless of αC-in or αC-out. Allosteric kinase inhibitors, competitive inhibitors that do not compete with ATP, bind to sites outside the kinase ATP-binding pocket. Depending on the location of the allosteric sites in relation to the ATP-binding pocket, these inhibitors are classified as Type III or Type IV. -R- usually refers to -NH, -SH, and -OH. Type V inhibitors, also known as covalent inhibitors, typically form a covalent bond with the thiol groups of cysteine or lysine residues at the ATP site. (**c**) Monoclonal antibodies (mAbs) primarily inhibit signal transduction through the binding of the antibody’s Fab segment to the receptor’s extracellular domain.

**Figure 2 ijms-25-05489-f002:**
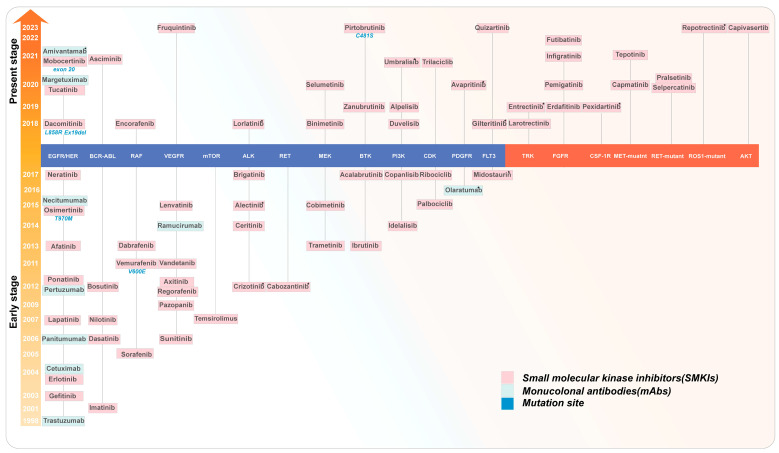
Timeline and target discovery history of FDA-approved small-molecular kinase inhibitors: 2018–1 March 2024. ‘*’ represents a variety of kinase inhibitors targeting multiple targets other than single-target subtypes and single-target mutation types. Multi-target inhibitors are classified based on the first FDA-approved target.

**Figure 3 ijms-25-05489-f003:**
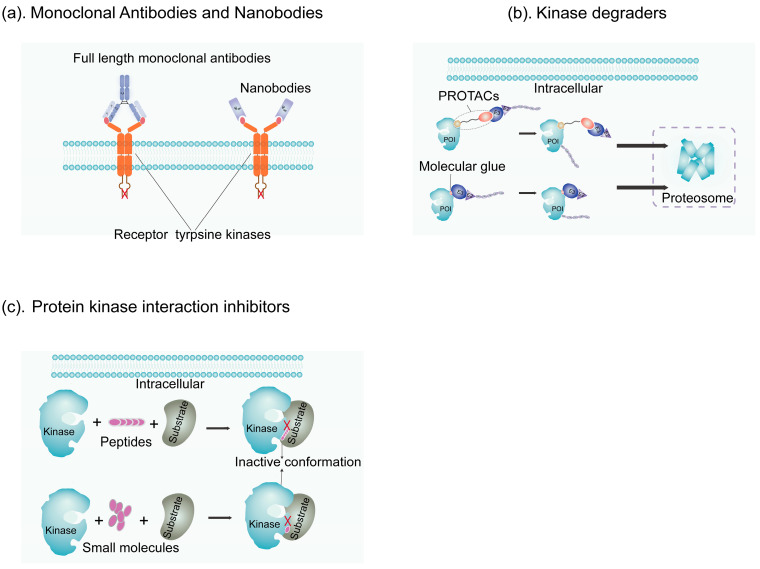
Mechanism of action of kinase-targeted cancer therapies, including monoclonal antibodies and nanobodies, kinase degraders, and protein–kinase interaction inhibitors. (**a**) Monoclonal antibodies (mAbs) primarily inhibit signal transduction through the binding of the antibody’s Fab segment to the receptor’s extracellular domain. Nanobodies lack the Fc portion, also known as single-domain antibodies or VHH. (**b**) PROTACs are bifunctional molecules that induce the proximity of E3 ligases and the protein of interest (POI) by linking them via ligands. Molecular glue is a monovalent molecule that binds to the surface of E3 ligase receptors, achieving binding to the target protein through protein–protein interactions and leading to the degradation of the POI. (**c**) Protein–kinase interaction inhibitors (PKIIs), including small molecules and linear peptides, inhibit the interaction between kinases and substrates, being ideal candidates for inhibiting protein–protein interactions (PPIs).

**Table 2 ijms-25-05489-t002:** Recent advances in antibodies and derivatives.

Targets	Compound Name	Alternatives	Drug Type	Indications in Research	Highest Study Phase
EGFR	Futuximab/Modotuximab	DS-992/DS-1024	Monoclonal antibody	Solid tumor, metastatic colorectal cancer	Phase 3 Company pipeline Symphogen A/S (Lyngby, Denmark)
HER2	HLX-22	AC-101, HLX-22	Monoclonal antibody	Gastric cancer, breast cancer, metastatic gastric cancer	Phase 3 Company pipeline AbClon, Inc. (Seoul, Republic of Korea).
EGFR	Pimurutamab		Monoclonal antibody	Cutaneous squamous cell carcinoma, squamous NSCLC	Phase 3 Company pipeline Shanghai Henlius Biotech, Inc. (Shanghai, China).
ROR1	Zilovertamab	Cirmtuzumab, UC-961, ZILO-301	Monoclonal antiboby	Mantle cell lymphoma, chronic lymphocytic leukemia	Phase 3 NCT05431179 University of California, (Los Angeles, CA, USA)
VEGFR2	Gentuximab	GenSci-043 GenSci043 GenSci043-01	Monoclonal antiboby	Advanced gastroesophageal junction adenocarcinoma, EGFR T790M mutation, NSCLC	Phase 3 CTR20220815 Changchun Genescience Pharmaceuticals Co., Ltd. (Changchun, China).
FGFR2	Bemarituzumab	FPA-144	Monoclonal antiboby	FGFR2b-positive gastroesophageal junction cancer	Phase 3 CTR20233946 Five Prime Therapeutics, Inc. (San Francisco, CA, USA).
IGF-1R	Ganitumab	AMG-479	Monoclonal antiboby	Bone metastases, Ewing’s sarcoma	Phase 3 NCT02306161 Amgen, Inc. (Thousand Oaks, CA, USA)
EGFR	JMT-101	JMT101	Monoclonal antiboby	Solid tumors, squamous non-small-cell lung cancer	Phase 2/3 NCT06319313 Shanghai JMT Biological Technology Co., Ltd. (Shanghai, China).
PDGFRβ	68Ga-BOT1712		Nanobody	Colon cancer, liver cirrhosis	Preclinical Company pipeline Cortalix BV (Groningen, The Netherlands)
HER2	RAD202		Nanobody	HER2-positive breast cancer	Preclinical Company pipeline Radiopharm Theranostics, Ltd. (Carlton, VIC, Australia).
PDGFRβ	177Lu-BOT1712		Nanobody	Colon cancer	Preclinical Company pipeline Cortalix BV

Clinical trials from https://clinicaltrials.gov/ and http://www.chinadrugtrials.org.cn/ accessed on 15 March 2024.

**Table 3 ijms-25-05489-t003:** New strategies for targeting kinases.

Target	Compound Name	Drug Type	Indications in Research	Highest Clinical Phase
NTRK	CG-001419	PROTACs	Solid tumors	Phase 1/2 Cullgen, Inc. (San Diego, CA, USA). CXHL2200331 *
EGFR	HSK40118	PROTACs	EGFR-positive NSCLC	Phase 1 Haisco Pharmaceutical Group Co., Ltd. (Chengdu, China). CTR20230926
GSPT1 × HER2	ORM-5029	PROTACs	HER2-positive solid tumors or breast cancer	Phase 1 Orum Therapeutics, Inc. (Daejeon, Republic of Korea). NCT05511844
EGFR L858R	CFT-8919	PROTACs	EGFR-positive NSCLC	Investigational new drug (IND) by FDA C4 Therapeutics, Inc. (Watertown, MA, USA). Company pipeline
BTK	BGB-16673	PROTACs	Chronic lymphocytic leukemia, B-cell lymphoma	Phase 1 BeiGene Ltd. (Cambridge, MA, USA). NCT05006716
BRAF, BTK	CFT-1946	PROTACs	BRAF V600 mutation-positive tumor, brain metastases, colorectal cancer	Phase 2 C4 Therapeutics, Inc. (Watertown, MA, USA). NCT05668585
BTK, BTK C481S	AC-0676	PROTACs	B-cell malignancies, tumors, chronic lymphocytic leukemia	Phase 1 Accutar Biotechnology, Inc. (Shanghai, China). NCT05780034
BTK	HSK-29116	PROTACs	B-cell lymphoma	Phase 1 Sichuan Haisco Pharmaceutical Co., Ltd. (Chengdu, China). NCT04861779
EGFR	HSK40118	PROTACs	EGFR-positive NSCLC	Phase 1 Haisco Pharmaceutical Group Co., Ltd. (Chengdu, China). CTR20230926
BTK	HZ-Q1070	PROTACs	B-cell malignancies	Phase 1 Hangzhou Hertz Pharmaceutical Co., Ltd. (Hangzhou, China). CTR20240471
BTK	UBX-303-1	PROTACs	Chronic lymphocytic leukemia	Clinical Application for approval Ubix Therapeutics Co., Ltd. (Seoul, Republic of Korea). Company pipeline
MEK, Raf	IK-595	Molecular glue	Colorectal cancer	Phase 1 Ikena Oncology, Inc. (Boston, MA, USA). NCT06270082
MAPK, Ras	NST-628	Molecular glue	Glioma, melanoma	Phase 1 Nested Therapeutics, Inc. (Cambridge, MA, USA). NCT06326411
CDK4, CDK6, c- Myc	A80.2 HCI	Molecular glue	Bladder cancer, breast cancer	Preclinical Suzhou Kintor Pharmaceuticals, Inc. (Suzhou, China). Company pipeline
c-Met	HIP-8	Polypeptide	Tumors	Preclinical Kanazawa University (Ishikawa, Japan) [147]

“*” Data from https://www.cde.org.cn/ accessed on 15 March 2024.

**Table 4 ijms-25-05489-t004:** Potential new targets for SMKIs.

Compound Name	Target	Mechanism of Action	Indications in Research	Clincial Phase	Clinical Trail
AL-8326	Aurora B, FGFRs, VEGFR	Aurora B inhibitors, FGFR antagonists, VEGFR antagonists	Small-cell lung cancer bile duct tumor, bladder cancer	Phase 3	CTR20233349
Chiauranib	Aurora B, CSF-1R	Aurora B inhibitor, CSF-1R antagonist, VEGFR junction inhibitor	Relapsing drug-resistant ovarian cancer, soft tissue sarcoma	Phase 3	Chiauranib
Tinengotinib	Aurora A, Aurora B, CSF-1R	Aurora A inhibitors, Aurora B inhibitors, CSF-1R antagonists	Bile duct epithelial carcinoma, triple-negative breast cancer	Phase 3	NCT05948475
CEP-11981	Tie-2, VEGF	Tie-2 antagonists	Adenosquamous carcinoma, gastrointestinal pancreatic neuroendocrine tumor	Phase 2	NCT05988918
Gandotinib	JAK2	JAK2 inhibitors	Blood tumor	Phase 2	NCT01594723
Itacitinib	JAK1	JAK1 inhibitor	Diffuse large B-cell lymphoma	Phase 2	NCT03139604
LY-2880070	CHK1	CHK1 inhibitors	Ewing sarcoma tumor	Phase 2	NCT05275426
Onvansertib	PLK1	PLK1 inhibitors	Small-cell lung cancer, recurrent metastatic colorectal cancer	Phase 2	NCT05450965
Prexasertib	CHK1 x CHK2	CHK1 inhibitors, Chk2 inhibitors	Metastatic triple-negative breast cancer	Phase 2	NCT05548296
Golidocitinib	JAK1	JAK1	Peripheral T-cell lymphoma, recurrent T-cell lymphoma	Phase 2	NCT04105010
Adavosertib	WEE1	WEE1 inhibitor	Local late-stage clear cell/renal cell carcinoma, metastatic renal cell carcinoma	Phase 2	NCT03284385
Azenosertib	WEE1	WEE1 inhibitor	Pancreatic ductal adenocarcinoma, acute myeloid leukemia	Phase 2	NCT06015659

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
