# Peer review of "Kinase Inhibitors and Kinase-Targeted Cancer Therapies: Recent Advances and Future Perspectives"

_ijms, 2024, doi:10.3390/ijms25105489_

Round 1

Reviewer 1 Report

Comments and Suggestions for Authors

This review aimed to provide a comprehensive summary of the mechanisms and targets of recently FDA-approved kinase inhibitors, as well as to discuss novel kinase inhibition strategies, including nanobodies, kinase degraders, and protein-kinase interaction inhibitors. Finally, the paper addressed the current challenges and future directions for the development of kinase inhibitors. The paper is interesting, supported by good figures and Tables, and puts all the mentioned issues together. However, there are a lot of other reviews containing the same information and are more complete, even in separate papers, thus the pertinence of this specific review is not clear. The novelty of the review should be also justified by the authors and emphasized in the manuscript.

Minor:

-Line 53-58: the sentence seems to be a repetition of the previous paragraph (Line 46-50)

-Line 61: 39 kinase inhibitors?

-Line 75: the sentence “New strategies…” should initiate a new paragraph

-Line 80: the sigla PPIs should be located before the word “inhibitors”

- Several typos throughout the manuscript, the document should be revised carefully

- The paper structure does not follow the IJMS guidelines, for example, the sections should be numbered:

1. Introduction

2.SMKIs

2.1. Types of SMKIs

2.1.1. ATP competitive inhibitors

(…)

Major:

1. Section 2.1 (Types of SMKIS): there are several papers describing and classifying the types of SMKIs, thus this section is unjustified in my opinion.

- Several typos in Figure 1: The examples of Type I inhibitors are the same as Type V, and Type III as Type IV. Substitute “et al.” by “etc”.

- Figure 1: the legend deserves a better description of the images.

2- Section 2.2 “FDA approved kinase inhibitors in oncology: 2018-2024” the authors should better explain why they chose the period between 2018-2024? Also, this section is included in section 2 “SMKIS”, but there are also mAbs included.

- Line 119 refers that in the period chosen by the paper it was approved 39 kinase inhibitors, however Figure 2 and Table 1 only contain 33 inhibitors.

- Line 220: The author referred to “SMKIs approved by the FDA between 2018 and March 2024.., however, both Table 1 and Figure 2 cite mAbs and thus are not only SMKIs. Also, in the description of Table 1 is cited “small molecular kinase inhibitors…” but it was also included mabs.

-Figure 2: Why is RET in the “orange” part of the figure if there are previous RET inhibitors approved (Cabozantinib)? If the justification is that because the newly approved drugs are for the mutant protein, EGFR and BTK have to be also included in the orange part of the arrow.

-Figure 2: “MERK” shouldn’t be MEK?

-Table 1 has too much information, the authors can make it simple by removing for example the “action mechanism” and maintaining the “Type”. In the column “First approval” the FDA word is unnecessary. Since they are already approved, the column “Alternatives” is also unnecessary.

3- Section 3 “New stratages of kinase targeted cancer therapies”

-Correct “Stratages” by Strategies

-Section 3.1 Monoclonal antibodies, should they be considered as a “new Strategy”?

-Table 2: Is easier for the reader if the table is organized by targets. The “mechanism of action” seems to be unnecessary. “highest study phase”, “clinical trial” and “company” should be summarized in only 1 column.

-Table 3: Same comment as Table 2-

-Figure 3: deserves a more descriptive legend.

4- Paper structure is confusing and not appealing, I suggest the following structure:

1. Introduction

2.Types of Kinase inhibitors

2.1. SMKIs

2.1.1. ATP competitive inhibitors

(…)

2.2. Monoclonal antibodies

(Figure 1: include examples of mAbs mechanism of action)

3. FDA-approved kinase inhibitors in oncology: 2018-2024

3.1. TRK

(…)

4. New strategies of kinase-targeted therapy

4.1. Monoclonal antibodies and derivatives

4.1.1. Monoclonal Antibodies

4.1.2. Nanobodies

4.2. Kinase degraders

4.2.1. PROTACs

(…)

4.3. Protein–kinase interaction inhibitors (PKIIs)

(…)

5. Future perspectives

5.1. Potential new targets

(…)

Comments on the Quality of English Language

Several typos throughout the manuscript, the document should be revised carefully.

Reviewer 2 Report

Comments and Suggestions for Authors

Kinase-targeted drug development strategies are shown to be important for cancer treatment. The review titled ‘Kinase Inhibitors and Kinase-Targeted Cancer Therapies: Recent Advances and Future Perspectives’ provides a comprehensive summary of recent FDA-approved kinase inhibitors based on their mechanisms of action and targets. The authors explored the emerging kinase inhibition strategies, such as nanobodies, kinase degraders, and protein-kinase interaction inhibitors and provided future perspectives in kinase inhibition based on various strategies. The challenges and future directions for the development of inhibitors are further reviewed in detail. As a researcher in the field of protein kinases, I very much appreciate the effort of the authors to provide such a detailed summary, which in particular will be useful for beginners in the kinase research field. 

Although the content is important I have a few comments (numbered 1-20) so that the review can be improved for better readability. 

  1. I recommend a native English speaker/writer to go through the content and adjust the sentences in a more appropriate manner. Often, certain sentences are confusing for the readers. For example lines 83- 84, line 521 etc.

  2. The authors claim that they provide a comprehensive summary of FDA-approved kinase inhibitors based on their mechanisms of action and targets. In my point of view it is incomplete, especially the mechanism of action is minimally mentioned. This can be elaborated, probably with a separate section of ‘mechanism of action’.

  3. The headings and subheadings should be improved, providing more details. While providing SMKIs as a main section heading, it is not clear what the authors are trying to say. It is important for the readers to get an overview of each section.

  4. Lines 131-132 and wherever mentioned: The DFG-in and out conformations are not always supporting whether it is active or inactive. This is still a debatable concept, and there are several other structural features to be considered.

  5. Page 3, lines 118-144: the authors are providing structural details of the kinase domain, however this is provided under the section SMKIs. Write a separate section of kinase domain structural features by providing a 3D structure as an additional figure.

  6. Be consistent in kinase domain terminology. Sometimes alpha-C and sometimes C are provided to mention alpha-C helix. I recommend writing alpha-C helix.

  7. Make sure all the contents in each table are cited with relevant references.

  8. Lines 46-48 is repeated in lines 54-56.

  9. It is better to avoid superscript to specify mutations. For example, line 66 and several other positions in the review.

  10. Lines 112 and 115: Use ‘this review’ instead of ‘this study’ and ‘the paper’.

  11. Figure 1: Provide a better, high resolution figure, with texts within the figure in larger font.

  12. Figure 1a: what is meant by ‘lope’?

  13. Wherever the small molecule inhibitors are listed (for example lines 157-158), provide the appropriate citations.

  14. Often certain abbreviations are not mentioned. For example BTK in line 203. I do not see the full form of BTK anywhere else in the review.

  15. Provide a list of abbreviations.

  16. All the SMKIs can be written in italic font or highlight as possible.

  17. Figure 2: it is of bad quality. Provide a high resolution figure.

  18. Line 214: rewrite the heading, include ‘years 2018-2024’.

  19. Line 228: After the section of ‘FDA approved kinase inhibitors in oncology: 2018-2024’ kinases such as TRK, FGFR etc are described without a proper transition. Please provide appropriate sentences and headings.

  20. The full form of kinases can be written on the headings with abbreviations in brackets.

  21. Figure 3: Use ‘mechanism of action’ instead of ‘action of mechanism’. 

These are the representative comments from selected sections and are applicable to the entire review. Review carefully and improve the content to address the general audience who may not be experts in kinase research.

Comments on the Quality of English Language

Please consult a native English speaker/writer to go through the content and adjust the sentences in a more appropriate manner. Often, certain sentences are confusing for the readers. The headings and subheadings are not organized well, the authors should definitely improve this. I added a comment on this in the report.

Reviewer 3 Report

Comments and Suggestions for Authors

The manuscript considers the current status in the development of kinase inhibitors. First, the authors provide the classification of small molecule kinase inhibitors. For each class, the general mechanism of action is described, and the list of inhibitors classified by the target is given. The extensive literature review has been performed on the subject, and the information is summarized in Table 1. Next, the authors describe available kinase inhibitors based on monoclonal antibodies and nanobodies, kinase degraders, and protein–kinase interaction inhibitors. Also, an extensive review of currently available kinase inhibitors is provided and summarized in Tables 2 and 3. Finally, in the last section authors provide the future perspectives on the development of the area, listing the possible new targets for kinase inhibitors, challenges to overcome, and technical advances that can facilitate development of kinase inhibitors. 

Overall, this is a very extensive review that can be a useful and valuable source of information for the specialists in the field. In addition to the list of kinase inhibitors, their targets, and types of action, authors provide discussions that can be helpful for development of new kinase inhibitors. The large amount of information provided in the review is ordered and presented in four tables. Three figures allow the reader to get the idea on the types of action of kinase inhibitors. A few minor comments are given below.

  • The first paragraph on page 2 (lines 46-52), and the first part of the second paragraph on page 2 (lines 53-58) describe the same information in different words.

  • Caption of Figure 3 does not allow the reader to completely understand the information presented in the figure, i.e. the caption lacks the short description of the mechanisms of action of kinase inhibitors. The main text also lacks a clear description on the mechanisms of action of kinase inhibitors other than SMKI.   

  • Resolution of figures is quite low

Round 2

Reviewer 1 Report

Comments and Suggestions for Authors

The article has improved significantly, but I still have minor revisions:

- Figure 1 still contains typos (selumetinib as an example for both type 3 and 4 inhibitors)

-Subtitle 2.1.5 "Type selection for the development of SMKIs" it's very confusing to me. Do the authors mean "Classification of inhibitor type in SMKIs development"?

-line 267 "Figure 2: Timeline of FDA approved kinase inhibitors: 2018-March 1, 2024" is to remove.

-Line 586: the figure legend should be only one and below the figure. 

-Table 3. "New strategies targeting kinase." shouldn't be "New strategies for targeting kinases"

Author Response

1. Summary

Thank you very much for taking the time to review this manuscript. Please find the detailed responses below and the corresponding revisions highlighted in red text in the re-submitted files.

Minor revisions:

Comment 1:

- Figure 1 still contains typos (selumetinib as an example for both type 3 and 4 inhibitors)

Response 1:

We feel sorry for our carelessness. In our resubmitted manuscript, the typo is revised. The example of type 4 inhibitors has been asciminib. Thanks for your correction.

Comment 2:

-Subtitle 2.1.5 "Type selection for the development of SMKIs" it's very confusing to me. Do the authors mean "Classification of inhibitor type in SMKIs development"?

Response 2:

In this section, we aim to explore the selection of kinase inhibitor types in the process of kinase inhibitor development. Specifically, we address the question of which type of kinase inhibitor to choose for development. Particularly noteworthy is the increasing prominence of covalent inhibitors in recent years, which have demonstrated enhanced specificity, potency, and safety. However, it is important to note that covalent inhibitors are not inherently superior to ATP competitive inhibitors, especially given the relatively advanced state of ATP competitive inhibitor development. Ultimately, we conclude that there is no definitive answer regarding which type of SMKIs currently holds the predominant advantage.

Comment 3:

-line 267 "Figure 2: Timeline of FDA approved kinase inhibitors: 2018-March 1, 2024" is to remove.

Response 3:

We apologize for this mistake. We have deleted line 267 [Figure 2: Timeline of FDA approved kinase inhibitors: 2018-March 1, 2024.]. Thank you for your careful review.

Comment 4:

-Line 586: the figure legend should be only one and below the figure.

Response 4:

We apologize for this mistake also. We have deleted line 586 [Figure 3. Mechanism of action of kinase targeted cancer therapies, including monoclonal anti-bodies and nanobodies, kinase degraders and protein kinase interaction inhibitor.]. And we have placed the Figure legend below. Thank you for your careful review.

Comment 5:

-Table 3. "New strategies targeting kinase." shouldn't be "New strategies for targeting kinases"

Response 5:

Thanks for your good suggestion again. We have modified [New strategies targeting kinase] to [New strategies for targeting kinases.]

Reviewer 2 Report

Comments and Suggestions for Authors

I appreciate the effort of the authors to incorporate my suggestions. 

By including the suggested modifications, the authors have addressed all my concerns and suggestions. This is now an excellent review on Kinase Inhibitors and Kinase-Targeted Cancer Therapies. I fully support the publication of this review in IJMS.

Author Response

Thank you very much for taking the time to review this manuscript.